# Tracking and perceiving diverse motion signals: Directional biases in human smooth pursuit and perception

**Xiuyun Wu**[1,2]*, **Miriam Spering**[1,2,3,4]

1 Graduate Program in Neuroscience, University of British Columbia, Vancouver, BC, Canada,
2 Department of Ophthalmology & Visual Sciences, University of British Columbia, Vancouver, BC, Canada,
3 Djavad Mowafaghian Center for Brain Health, University of British Columbia, Vancouver, BC, Canada,
4 Institute for Computing, Information and Cognitive Systems, University of British Columbia, Vancouver, BC, Canada

* xiuyunwu@student.ubc.ca

**Data Availability Statement:** Materials including data, experiment codes, and analysis codes of this manuscript are available on Open Science Framework (DOI: 10.17605/OSF.IO/7BC8J).

## Abstract

Human smooth pursuit eye movements and motion perception behave similarly when observers track and judge the motion of simple objects, such as dots. But moving objects in our natural environment are complex and contain internal motion. We ask how pursuit and perception integrate the motion of objects with motion that is internal to the object. Observers ($n = 20$) tracked a moving random-dot kinematogram with their eyes and reported the object's perceived direction. Objects moved horizontally with vertical shifts of 0, ±3, ±6, or ±9° and contained internal dots that were static or moved ±90° up/down. Results show that whereas pursuit direction was consistently biased in the direction of the internal dot motion, perceptual biases differed between observers. Interestingly, the perceptual bias was related to the magnitude of the pursuit bias ($r = 0.75$): perceptual and pursuit biases were directionally aligned in observers that showed a large pursuit bias, but went in opposite directions in observers with a smaller pursuit bias. Dissociations between perception and pursuit might reflect different functional demands of the two systems. Pursuit integrates all available motion signals in order to maximize the ability to monitor and collect information from the whole scene. Perception needs to recognize and classify visual information, thus segregating the target from its context. Ambiguity in whether internal motion is part of the scene or contributes to object motion might have resulted in individual differences in perception. The perception-pursuit correlation suggests shared early-stage motion processing or perception-pursuit interactions.

## Introduction

Living in a dynamic visual environment requires awareness of moving objects around us and the ability to rapidly evaluate and judge their direction and speed. In order to keep track of the moving objects around us, humans use smooth pursuit eye movements. This type of smooth, slow rotation of the eyes allows the continuous projection of an object of interest close to the

**Funding:** This work is supported by a UBC Four-Year fellowship to X.W., and a Natural Sciences and Engineering Research Council of Canada Discovery Grant and Accelerator Supplement to M. S. The funders had no role in study design, data collection and analysis, decision to publish, or preparation of the manuscript.

**Competing interests:** The authors have declared that no competing interests exist.

fovea, the small area on our retinae that enables high visual acuity [1]. Even though humans and other primates can make different kinds of targeted eye movements that aid perception [2,3]—such as saccades (quick jumps of the eyes to interesting objects) and fixation (holding the eyes within a small spatial range)—smooth pursuit is special in several ways. This eye movement has almost exclusively evolved in primates [4], and is so tightly linked to visual motion that it can only be initiated if the observer perceives motion [5]. It is difficult to initiate a smooth response to non-visual motion signals, such as during auditory, tactile, or somato-sensory stimulation [6]. Whereas illusory motion can be tracked [7,8], pursuit in the complete absence of a stimulus is not usually possible.

It appears to follow logically that this type of eye movement should be tightly linked to how we perceive visual motion. Indeed, several studies have shown an association between both responses (for reviews, see [9,10]). For instance, both pursuit and perception show similar accuracy and variability in direction and speed discrimination tasks [11–14]. These perception-action associations are usually described under laboratory conditions that involve sparse visual stimuli, such as the translating motion of a single dot or static pattern [11–14]. However, in daily life we usually encounter moving objects that contain additional internal motion signals (Fig 1). For example, when we observe a swarm of insects, each insect inside the swarm has its own individual motion direction, yet the swarm also follows an overall (global) motion direction (Fig 1A). Individual motion and global object motion occur on different spatial scales. Even for a single object, such as a flying and rotating volleyball (Fig 1B), object components (ball stripes) might present different motion signals from the global object motion.

Here we investigate whether the assumed association between perception and pursuit extends to situations in which potentially diverse visual motion signals have to be integrated across space. Specifically, we ask whether the internal motion of an object affects how we perceive and track its trajectory. This question is important because the integration of motion signals across space is a necessary prerequisite in order to derive an object's veridical motion for accurate perception and pursuit in a more complex natural environment.

Whether this signal integration results in similar or different behavioral outcomes of perception and pursuit appears to depend on the stimulus environment and task. Evidence for perception-pursuit associations comes from studies employing objects that contain ambiguous local (internal) motion signals. A typical example is the "aperture problem", in which the object's motion can only be decided if it is revealed or perceived in its entirety. Imagine a

a                                                          b

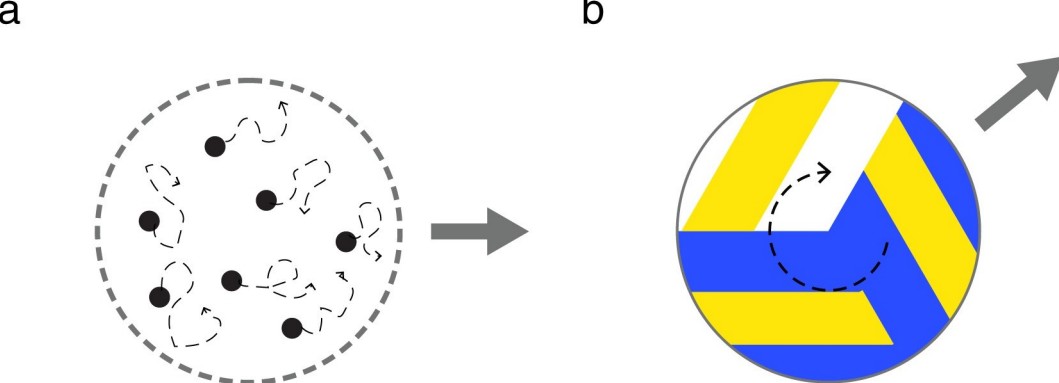

**Fig 1. Illustration of motion signals on different spatial scales.** (a) Simple illustration of internal and object motion as observed in a swarm of insects. (b) Flying and rotating volleyball. The grey solid arrow indicates the object motion which operates on a larger spatial scale, and the black dashed arrows indicate internal motion within each object, which is different from the object motion and operates on a smaller spatial scale.

diagonally tilted bar translating horizontally and viewed through an aperture that covers the ends of the bar. Due to its orientation, the translation could be seen as either diagonal or horizontal. The ambiguity can only be resolved when the whole bar is visible. In this case, motion signals from local edges, such as the middle part of the stimulus, have to be combined with motion of structures, such as the ends of the stimulus, to inform its veridical direction. Perception and pursuit show a similar temporal development of directional biases in response to the aperture problem: perceived direction [15] and smooth pursuit direction [16–18] are initially both biased to the local motion direction. These biases are then corrected toward the object motion direction after further integration of motion signals. When viewing segments of line-figure objects behind apertures (e.g., only the middle part of the four edges of a diamond shape is visible), direction perception and pursuit of the object motion are affected similarly by whether apertures are visible or not, which affects the degree to which an object is perceived as a whole rather than as individual line segments [19,20]. In sum, perception and pursuit seem to follow the same representation of object motion.

Evidence for perception-pursuit dissociations in motion signal integration comes from studies that require the assessment of the relative relationship between different motion signals. Motion signals at different spatial scales may not only come from within the object, but can also be associated with a small object moving against a large background. For example, when observers were asked to judge the speed of a small target in the presence of a large moving background, the perceived velocity of the target increased with decreasing context velocity [21]. By contrast, pursuit velocity increased with increasing background speed [21–24]. When the different motion signals are both within the target (a translating and drifting grating), a dissociation was found between speed perception and pursuit velocity [25]. No consistent bias was found in perceived speed, whereas pursuit velocity followed the average speed of the translating and drifting motion.

In sum, when observers judge an object's direction by integrating diverse motion signals within an object, perception and pursuit tend to be associated. Both are affected by object and internal local motion. By contrast, when observers judge an object's speed when the relationship of the diverse motion signals (within an object or between an object and its surround) needs to be determined, perception and pursuit tend to be dissociated. Pursuit is always affected by all diverse motion signals, whereas perception is either affected in the opposite direction to pursuit [21], or is not affected consistently [25].

In the current study we focus on object direction and investigate how the object's internal motion direction and its global motion direction are integrated to drive perception and pursuit of global object motion. Human observers viewed a random-dot kinematogram (RDK) translating across the screen while constant dot motion inside the RDK was shown. We expect that pursuit will be biased toward the internal motion direction, and that it will reflect average direction signals from both sources (motion assimilation). Congruent with studies that report perception-pursuit associations [19,20], perception could be similarly biased by internal motion direction as pursuit, and follow a motion assimilation effect. This finding would imply that perception and pursuit use the same strategy to integrate object and internal motion. Alternatively, perception could follow a motion contrast effect (bias against the internal motion direction), as described in studies finding dissociations [21]. A dissociation between perception and pursuit would indicate that different strategies are used by the two systems to process object and internal motion. We further examine whether biases in perception and pursuit are correlated across observers and within observers, which would indicate to what extent both responses are related. Whereas an across-observer correlation indicates an overall link between perception and pursuit, a trial-by-trial correlation indicates that they also share noise sources that commonly affect variability in sensorimotor responses [12,14]. Our results will

further our understanding of how perception and pursuit are related under more complex and ecologically valid situations.

## Methods

### Observers

We recruited 20 observers (age $M$ = 26.75, $std$ = 3.42 years; nine female and 11 male) with self-reported normal or corrected-to-normal visual acuity and no history of ophthalmic, neurologic, or psychiatric disease. The sample size was determined by an a priori power analysis using G*Power [26], and was suggested to be sufficient to detect an effect size of Cohen's $f$ = 0.25 and power of 0.85 for statistical tests (ANOVA) conducted to test our hypotheses. A medium effect size of 0.25 [27] was chosen since Hughes [25] found a large effect on pursuit (Cohen's $f$ = 2.21, calculated based on Equation 13 from [28]), but perception showed no effect. The University of British Columbia Behavioral Research Ethics Board approved all experimental procedures, and all observers participated after giving written informed consent. Observers received $10 CAD remuneration for each hour they participated in the experiment.

### Visual stimuli and setup

Stimuli were random-dot kinematograms (RDK) presented in an aperture with a radius of 1˚. Each RDK consisted of 31 uniformly distributed white dots at a density of 10 dots/deg$^2$ and presented at a luminance of 52.86 cd/m$^2$ on a grey background (13.87 cd/m$^2$). The whole RDK (aperture + dots) moved across a computer screen at a constant speed of 10˚/s. We refer to this motion as the object motion. The object moved horizontally to the right, and at one of seven angles relative to horizontal, either at no deviation (0˚) or oriented downwards (-9˚, -6˚, -3˚) or upwards (3˚, 6˚, or 9˚; see Fig 2B). Dots shown within the RDK were either stationary (baseline), or moved coherently at a constant speed of 5˚/s in the upward (90˚) or downward (-90˚) direction relative to the aperture (Fig 2B). The absolute dot velocity on the screen depended on the angle between the object and internal motion directions. The spatial displacement (Δx) per frame was 7 to 8 arcmin (4 to 5 pixels). The duration of each frame (Δt) was approximately 12 ms, at the rate equal to the screen refresh rate of 85 Hz. Dot lifetime was nine frames

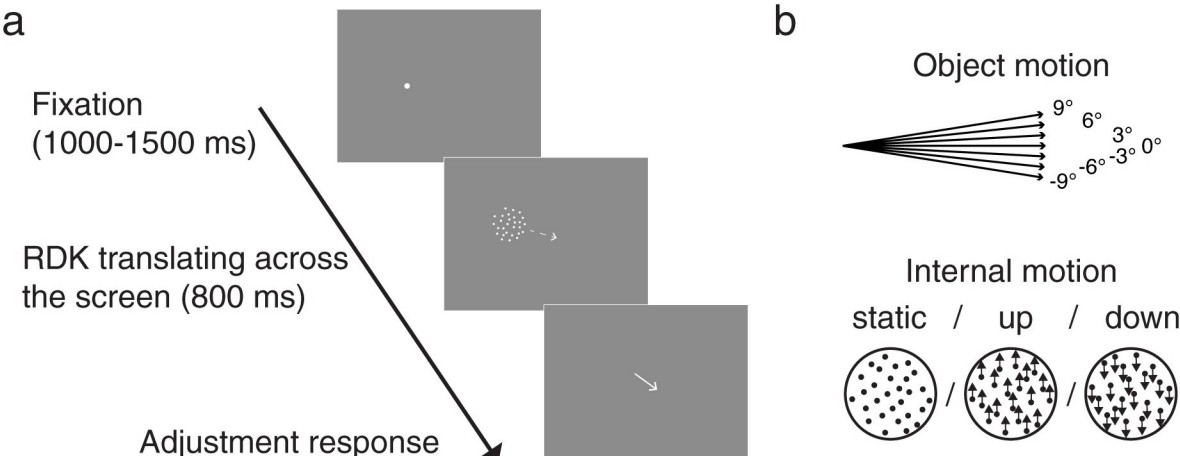

**Fig 2.** Trial timeline (a) and object and internal motion direction (b). At the beginning of each trial, a fixation point was shown for 1000–1500 ms, followed by the RDK translating across the screen for 800 ms (dashed arrow is for illustration purposes and was not shown in the experiment). A response arrow was shown after the RDK until observers clicked the mouse button to submit their response.

(approximately 106 ms). Each dot was initialized with a random duration of lifetime left, in order to avoid a whole-RDK flash when replacing the expired dots. At the end of its lifetime, a given dot reappeared at a random location within the aperture. When a dot moved out of the aperture, it re-entered from the opposite side of the aperture.

Observers were seated in a dimly-lit room and viewed all stimuli on a gamma-corrected 36.7 cm × 27.5 cm CRT monitor (ViewSonic G90fB; resolution 1280 × 1024 pixel; refresh rate 85 Hz). The viewing distance was 55 cm. Each observer's head was stabilized using a chin-and-forehead-rest. Stimuli and procedure were programmed in MATLAB R2019a (The Math-Works Inc., Natick, MA) and Psychtoolbox Version 3.0.12 [29–31].

## Procedure and design

Each block of trials started with a standard 5-point calibration of the eye tracker. Each trial then started with a white fixation point which observers were asked to fixate until it disappeared after 1000–1500 ms (Fig 2A). Fixation was monitored online: if eye position was further than 1˚ from the center of the fixation point, the fixation point turned red and the countdown of fixation duration was paused until the observer regained accurate fixation. After fixation, we used an adapted step-ramp paradigm: the RDK appeared slightly backward on its motion trajectory to the left of the fixation location, and then moved rightward along its motion trajectory. Such a step-ramp paradigm is typically used to prevent the initial use of saccades and instead elicit a smooth initiation of pursuit to the target [32]. The RDK moved across the screen for 800 ms, and observers were asked to follow the object motion as smoothly as possible with their eyes. After the RDK disappeared, a white response arrow appeared at the screen center. Observers were instructed to report the object motion direction by adjusting the angle of the arrow using the track-ball mouse (the arrow would always point to the cursor position) and then click the mouse button to submit the response. There was no time pressure for responding, and no feedback was given. To encourage observers to not use location as a cue by simply pointing the arrow to the final location of the RDK, the object motion trajectory contained spatial jitter. The middle (half-way) point of the object motion trajectory was by default at the screen center; the addition of the spatial jitter resulted in a random placement of the half-way point within 1˚ of the screen center. This spatial randomization would result in wrong answers if simply pointing to the last location of the RDK, which was explained to the observers. They were strongly encouraged to focus on the angle of the object motion trajectory instead of using the last location of the RDK as a reference.

The manipulations of object and internal motion result in a 7 (object motion) × 3 (internal motion) within-subject design. Each experimental block consisted of 42 trials, with two trials per condition, all randomly interleaved. Observers first performed one practice block of 21 trials (one trial per condition) to get familiar with the task, and then completed 10 experimental blocks. Data from the practice block were not analyzed. Including breaks between blocks, the experiment took between 50 and 75 minutes.

## Eye movement recording and analysis

The position of the right eye was recorded at a sampling rate of 1000 Hz using a video-based eye tracker (EyeLink 1000 tower-mount, SR Research Ltd., Kanata, ON, Canada). Eye movements were then analyzed offline using custom-made MATLAB functions. Eye position, velocity, and acceleration data were filtered with a second-order Butterworth filter (cutoff frequencies of 15 Hz for position and 30 Hz for velocity and acceleration). Eye direction was calculated as the angle of the velocity vector, represented in the same way as object motion direction (horizontal right is 0˚, and counterclockwise is positive). Trials with blinks during

RDK presentation or trials with signal loss were manually labeled as invalid and excluded (5% on average across observers). We also excluded trials in which the perceptual report was to the left ($<0.2\%$ across observers).

Saccades were detected based on an acceleration criterion: the acceleration trace was segmented by zero-crossing points, and peak acceleration within each segment was calculated. If at least two consecutive segments had absolute peak acceleration larger than $400°/s^2$, these segments were defined as saccades. An acceleration threshold was used to accurately detect saccades of small amplitude and velocity. Saccade detection was confirmed by visual inspection of the velocity traces in each trial. Saccades were excluded from the analysis of smooth pursuit. On average, observers made saccades in $M = 48 \pm 20\%$ (SD) of the trials. In trials with saccades, observers made $M = 1.09 \pm 0.10$ (SD) saccades per trial.

Smooth pursuit onset was detected in each trial by fitting a piecewise linear function with two segments and a break point (as previously reported in [33]) to the filtered and saccade-interpolated velocity traces (eye positions were linearly interpolated between saccade onsets and offsets). The fitting window to detect pursuit onset started at RDK onset, and the end of the window was 150 ms after the point at which eye velocity consistently exceeded four times the standard deviation of the fixation noise [34]. We calculated the averaged pursuit direction during the steady-state phase, defined as the time period between 140 ms after pursuit onset to 100 ms before the end of the visual motion display. We excluded the last 100 ms due to known anticipatory slowing of the eyes.

## Hypotheses and statistical analysis

We aim to test the following hypotheses. First, we expect that pursuit direction will be biased toward the internal motion direction, showing a bias in the average motion direction (motion assimilation; Fig 3A). Second, we expect that perceived direction will be affected by internal motion. It could either show a similar assimilation bias as pursuit (Fig 3A), or alternatively, it could be biased against the internal motion direction (motion contrast; Fig 3B). To test these hypotheses, we calculated directional bias as the difference between tracked / perceived

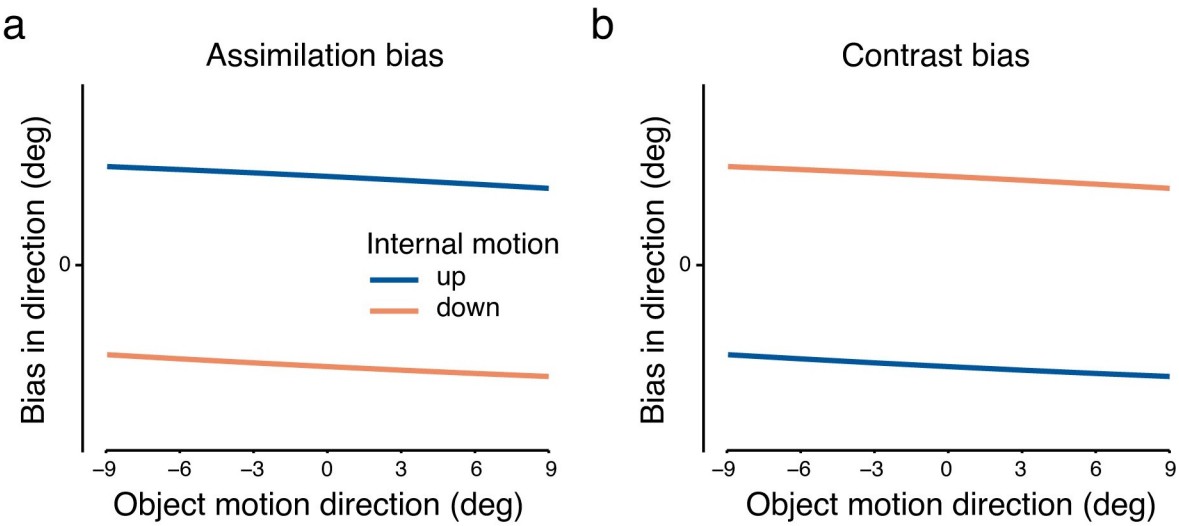

**Fig 3.** Alternative hypotheses for (a) assimilation bias and (b) contrast bias. The plots show the pattern of averaging/contrasting object and internal motion directions.

direction in the internal-motion conditions and the baseline condition with no internal motion (static dots inside the RDK). To assess whether perception and pursuit followed motion assimilation—the average between the internal and the object motion directions—or contrast—the difference between internal and object motion directions, we used two-way repeated measures of analysis of variances (rmANOVA) with *internal motion* (upward/downward) and *object motion* (seven angles) as factors. Two rmANOVAs were performed, one with pursuit direction bias as dependent variable, and the other with perceptual direction bias as dependent variable. A significant main effect of *internal motion* would indicate a bias. The direction of this main effect would then indicate the type of bias: an assimilation bias describes a bias in the same direction as internal motion, i.e., an upward trend in perception and pursuit in response to upward internal motion and a downward trend in response to downward internal motion (Fig 3A), whereas a contrast bias is indicated by the opposite pattern (Fig 3B). A significant main effect of *object motion* is expected because averaging or contrasting object and internal motion would result in the bias values being more negative when object motion direction is more positive (negative slopes in Fig 3). We do not expect any significant interaction effects of *internal motion × object motion*, although one could be observed if perception or pursuit responded stronger to either upward or downward motion (e.g., [35]).

If perception and pursuit biases were related, we would expect to see a significant across-observer correlation. To examine this, we calculated the Pearson's correlation coefficient between directional biases in perception and pursuit. For each observer, we averaged the bias across conditions so that all trials have an upward internal motion direction (a positive value would indicate bias in the same direction as internal motion). Further, if perception and pursuit shared noise sources during motion processing, we would expect to see a within-observer trial-by-trial correlation between perceptual and pursuit biases. To examine this, we fitted a linear mixed-effects model of perceptual bias with *object motion*, *internal motion*, and *pursuit bias* as fixed effects, together with all possible interaction effects and individual intercept and slope as random effects [formula: perceptual bias ~ object motion + internal motion + pursuit bias + object motion × pursuit bias + internal motion × pursuit bias + object motion × internal motion + object motion × internal motion × pursuit bias + (1+ pursuit bias | observer)]. A significant fixed effect of *pursuit bias* would indicate a significant trial-by-trial correlation.

We used an α-error probability of 5% ($p < 0.05$ is considered significant). We applied Mauchly's test for sphericity for factors having more than two levels in rmANOVAs. If the assumption of sphericity is violated, we reported Greenhouse-Geisser corrected $p$ values ($p$ [GG]) when the violation is more severe (epsilon < 0.75), or Huynh-Feldt corrected $p$ values ($p$[HF]) when the violation is less severe (epsilon > = 0.75). For effect sizes, we report partial eta-squared ($\eta_p^2$) in two-way ANOVAs. The statistical tests were conducted in R Version 4.1.2 (package "lme4", [36]; package "lmerTest", [37]; package "ez", [38]; package "psychReport", [39,40]) and MATLAB R2020a.

## Results

### Veridical perception and tracking of object motion in the absence of internal motion

In this study, we compared perception and pursuit in trials in which an object's motion direction was either combined or not combined with internal motion. Fig 4A shows an example eye position trace from one single trial for a representative subject in comparison to the target trajectory, and Fig 4B shows average eye position traces across observers in each object motion condition. Pursuit was initiated 132 ms (*SD* = 16 ms) after target onset on average, and then followed the veridical object motion direction in trials in which there was no internal motion.

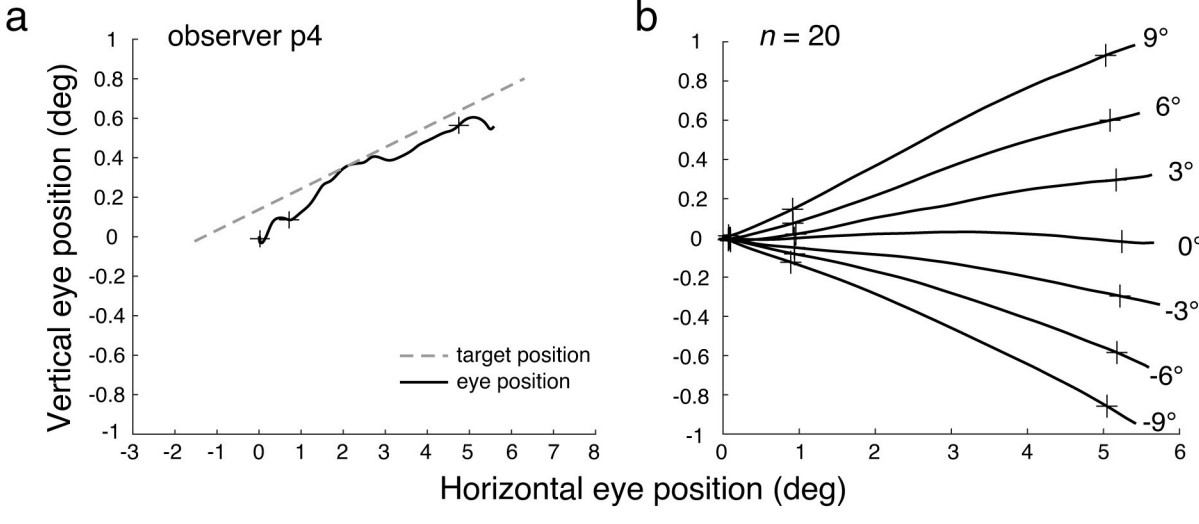

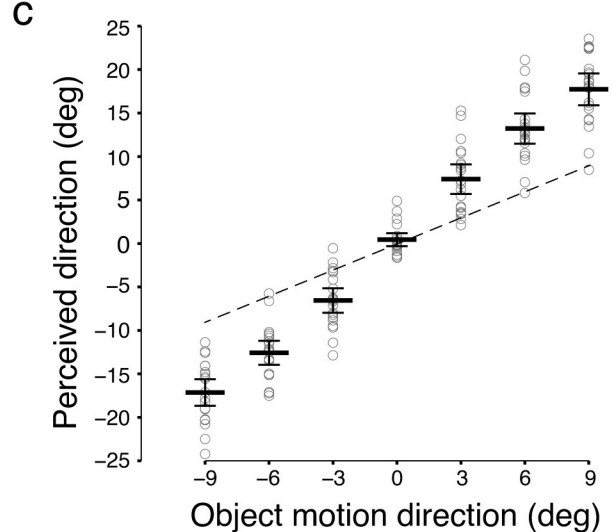

**Fig 4. Eye position traces and perceived direction in the baseline internal motion condition.** (a) Example eye position trace from one trial without internal motion from RDK onset to RDK offset. The object motion direction (dashed line indicates trajectory of RDK center) is 6˚. From left to right, the crosses indicate time points of pursuit onset, and the start and end of steady-state phase analysis window, respectively. (b) Average eye position traces across observers in the baseline internal motion condition. All traces are aligned to start from RDK onset at location (0, 0), and end at RDK offset. Numbers denote the object motion directions. (c) Perceived direction in the baseline condition averaged across observers ($n$ = 20). Horizontal bars indicate mean perceived direction across observers, and error bars indicate the 95% CI. Circles indicate the mean perceived direction of individual observers. The dashed line indicates the identity line. If the perceived direction was accurate without overestimation, data should fall on the identity line. CI, confidence interval.

Perceived direction roughly matched the object motion direction, with a general trend to over-estimate (Fig 4C). To examine the effect of internal motion direction on perceived and tracked object motion, we subtracted the baseline responses to calculate the perceptual and pursuit biases in trials with internal motion (upward/downward) and examined how the biases were affected by internal motion directions.

## Assimilation bias in smooth pursuit direction

In trials in which object motion was combined with diverse internal motion signals (either upward or downward relative to horizontal), we observed that pursuit eye movements were

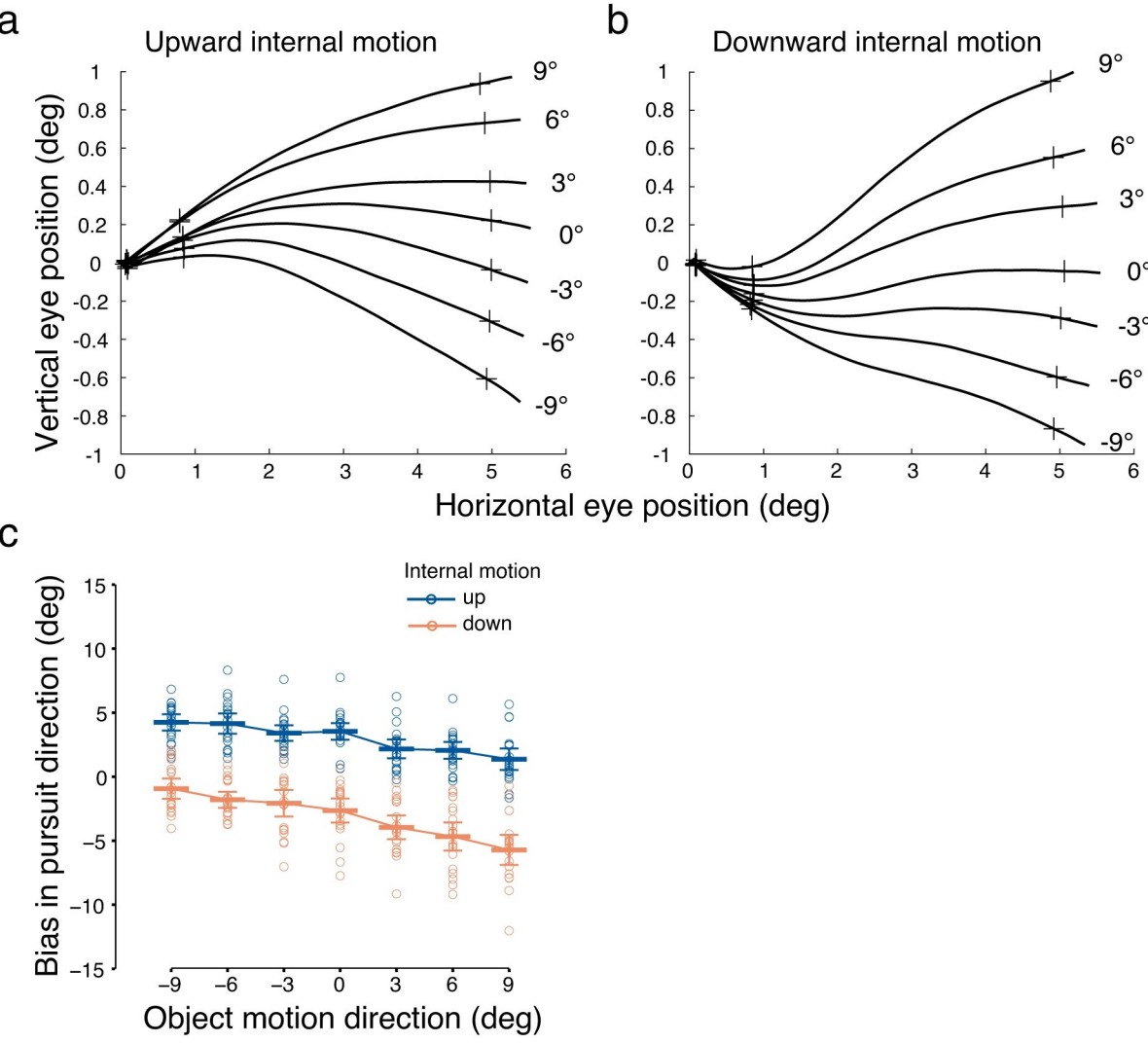

**Fig 5. Eye position traces and pursuit direction biases in conditions with internal motion.** Average ($n$ = 20) eye position traces for (a) upward and (b) downward internal motion. The crosses indicate time points of pursuit onset, and the start and end of the steady-state phase analysis window. Numbers denote the object motion directions. (c) Bias in pursuit direction averaged across all observers ($n$ = 20). Horizontal bars indicate the mean across observers, and error bars indicate the 95% CI. The circles indicate the mean of individual observers.

biased toward the internal motion direction (Fig 5A and 5B). When internal dots moved upward, the eye was biased in the upward motion direction, and vice versa for downward motion. This assimilation bias was confirmed by a significant main effect of *internal motion* on pursuit ($F(1,19)$ = 167.88, $p < 0.001$, $\eta_p^2$ = 0.90), with positive biases in the upward internal motion condition and negative biases in the downward motion direction (Fig 5C). The assimilation bias pattern in pursuit was observed in all observers, as can be inferred from individual data points in Fig 5C, which fall into two separate clusters around the mean per internal motion direction. We also found a negative slope of pursuit bias across object motion conditions, shown by a significant main effect of *object motion* ($F(6,114)$ = 28.07, $p < 0.001$, $\eta_p^2$ = 0.60). Moreover, we observed an asymmetry in pursuit bias with a larger bias in response to downward vs. upward internal motion direction. This observation was confirmed by a significant *object motion* × *internal motion* interaction ($F(2.94,55.80)$ = 4.27, $p$[GG] = 0.01, $\eta_p^2$ =

0.18). In summary, pursuit eye movements are biased toward the internal motion direction, following a motion assimilation effect, consistent with our hypothesis (Fig 3A).

### Inconsistent bias in perceived direction

Observers were asked to report the perceived direction of the RDK by adjusting a response arrow in the perceived direction. They did so with an average reaction time of 2.13 ± 0.66 (mean ± SD) seconds. In contrast to the consistent bias in the average motion direction in pursuit, perceived direction across all observers seemed to be not affected by internal motion direction (Fig 6A). Congruently, the main effect of *internal motion* on perception was not significant ($F(1,19) = 0.57$, $p = 0.46$, $\eta_p^2 = 0.03$). The negative slope of biases across object motion conditions (Fig 3) was still observed, shown by a significant main effect of *object motion* ($F$

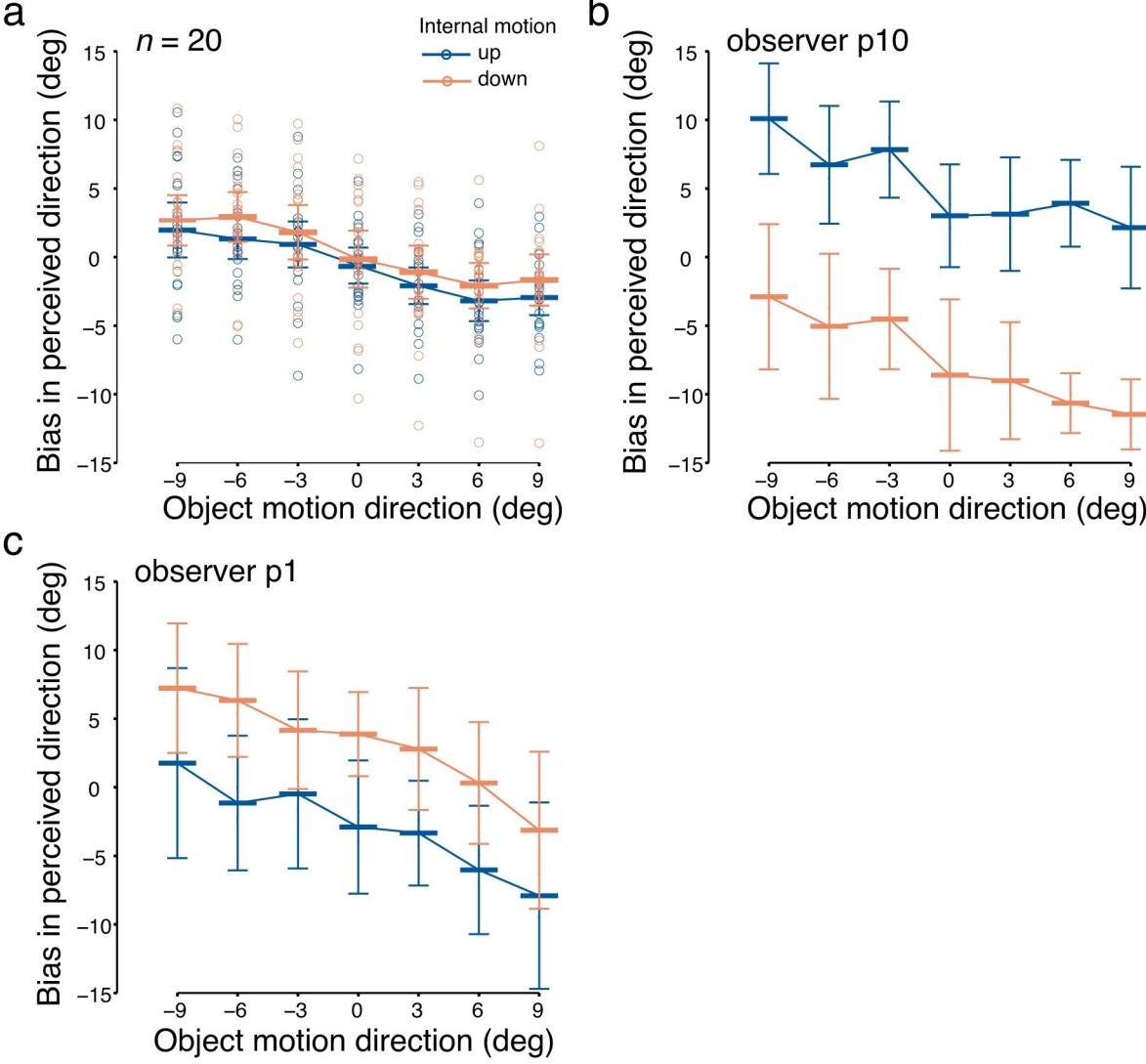

**Fig 6. Biases in perceived direction in conditions with internal motion.** (a) Bias in perceived direction averaged across all observers. Horizontal bars indicate the mean across observers, and error bars indicate the 95% CI. The following panels show examples of individual data showing (b) assimilation bias or (c) contrast bias in perception. Horizontal bars indicate the mean perceived direction bias across trials, and error bars indicate the standard deviation.

$(3.57, 67.74) = 23.12$, $p[GG] < 0.001$, $\eta_p^2 = 0.55$), but there was no up/down asymmetry in perceptual biases, indicated by a non-significant *object motion × internal motion* interaction ($F$ $(2.81, 53.43) = 0.39$, $p[GG] = 0.74$, $\eta_p^2 = 0.02$). In sum, our findings do not support a perceptual bias in either the average (assimilation) or relative (contrast) motion direction.

Further examination of individual data revealed a possible reason for the lack of perceptual bias in the group data. Whereas some participants showed an assimilation bias (Fig 6B), other participants' perception was biased in the contrast motion direction (Fig 6C). This may also explain why the negative slope of perceptual bias across object motion directions is preserved when averaging across observers, because the slope is similar in both assimilation and contrast biases. Detailed individual data and different perceptual bias patterns are described in the section on Exploratory Analyses (see below).

## Correlation between perception and pursuit biases

Results above show that whereas the tracking direction of the RDK in all observers was biased toward the internal motion direction, perceived direction was not biased on average. However, perception and pursuit could still share motion processing at a low level, and further integrate motion signals in different ways. If motion processing is shared, we should see correlations between perceptual and pursuit biases, regardless of their diverse bias patterns. We conducted both across-observer and trial-by-trial correlations between perceptual and pursuit biases, to examine to what extent the two systems are related.

We found that even though perception and pursuit did not show similar response bias patterns, their directional responses were related, as indicated by a significant across-observer correlation ($r = 0.75$, 95% CI = [0.45, 0.89], $p < 0.001$; Fig 7). These findings imply that participants with a larger pursuit bias in the direction of internal motion tended to also have a larger perceptual bias in the direction of internal motion. By contrast, participants with a smaller pursuit bias tended to exhibit a perceptual bias toward the motion contrast direction.

To further examine whether internal noise in the motion processing systems is shared between perception and pursuit, as would be indicated by significant trial-by-trial correlations which signify that the variability of the responses is related, we fitted a linear mixed effects model. We did not find significant trial-by-trial correlations, indicated by a non-significant fixed effect of pursuit bias on perceptual bias (estimate = 0.07, 95% CI = [-0.08, 0.23], $t(23.52)$ = 0.90, $p = 0.38$). All other effects were non-significant, except the fixed effects of object motion (estimate = -0.29, 95% CI = [-0.34, -0.24], $t(5218) = -12.13$, $p < 0.001$) and internal motion (estimate = -1.8, 95% CI = [-2.21, -1.39], $t(5232) = -8.61$, $p < 0.001$). This suggests that perception and pursuit are unlikely to share sensory or motor noise sources in this particular task.

Overall, our results point at a dissociation in response patterns between perception and pursuit. Whereas the perceptual bias was inconsistent across observers, pursuit was consistently biased toward the internal motion direction. Notwithstanding these differences, perceptual and pursuit biases were correlated across observers. Participants with larger assimilation bias in pursuit tended to have an assimilation bias in perception, whereas participants with smaller pursuit biases tended to exhibit a contrast bias in perception. In the following section, we will report exploratory analyses to further examine potential subgroups of participants with different perceptual biases.

## Exploratory analyses on individual differences

We hypothesized that perception would either show an assimilation bias similar to pursuit, or a contrast bias different from pursuit. However, the overall perceptual bias across observers showed no effect of internal motion (Fig 6A). Individual data implied that people might differ

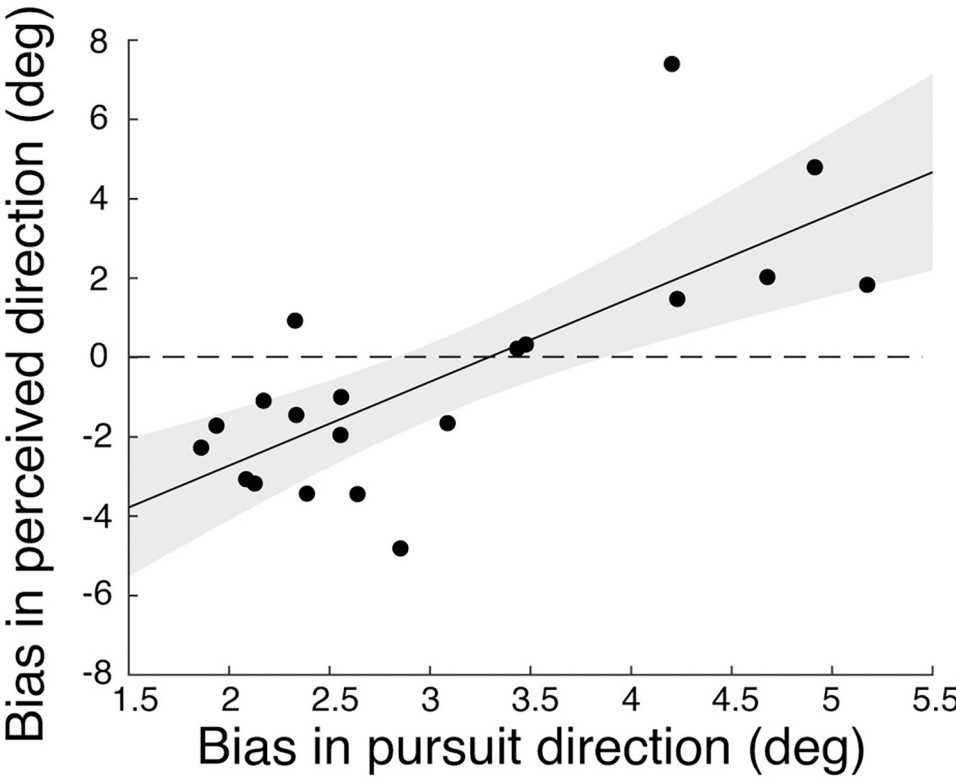

**Fig 7. Correlation between the bias in perceived direction and the bias in pursuit direction across observers (*n* = 20).** Each data point indicates the average value between internal motion conditions for one observer (positive values indicated bias in the direction of internal motion). The solid line shows the least squares fit. The shaded area indicates the 95% CI. The dashed line indicates zero perceptual bias.

in how their perception was biased (Fig 6B and 6C), thus resulting in an overall null result. The scatterplot in Fig 7 shows that the majority of data points falls in the lower left quadrant, with a relatively small pursuit bias and a contrast bias in perception. However, some data points fall in the upper right quadrant, revealing a strong pursuit bias and an assimilation bias in perception that is congruent with the pursuit result; other data points cannot be classified into either of these groups.

To investigate this clustering, we performed a latent profile analysis on perceptual bias across observers. Latent profile analysis uses maximum likelihood estimation to identify heterogenous subgroups with different patterns of variables measured [41–43]. In our case, the measured variables are the perceptual bias across object motion conditions, and the patterns are assimilation vs. contrast. We calculated the bootstrapped mean (of 1000 simulations) for each *object motion × internal motion* condition in each observer, then averaged the perceptual bias between internal motion conditions (conditions were merged so that internal motion was upward in all trials). Each observer had seven perceptual bias values, one for each object motion condition. Assimilation bias corresponds to positive values, and contrast bias corresponds to negative values. A latent profile analysis was performed with the assumption that observers came from two subgroups (analysis was done using R package "tidyLPA", [44]). The latent profile analysis does not prove the existence of subgroups, but allows us to explore the results if this assumption was true.

The algorithm was able to classify observers into two groups reliably, indicated by high probability of group classification (*M* = 98.91%, *SD* = 1.63%). Fig 8 shows perceptual bias patterns for the two subgroups, an assimilation group (*n* = 6) and a contrast group (*n* = 14). Each

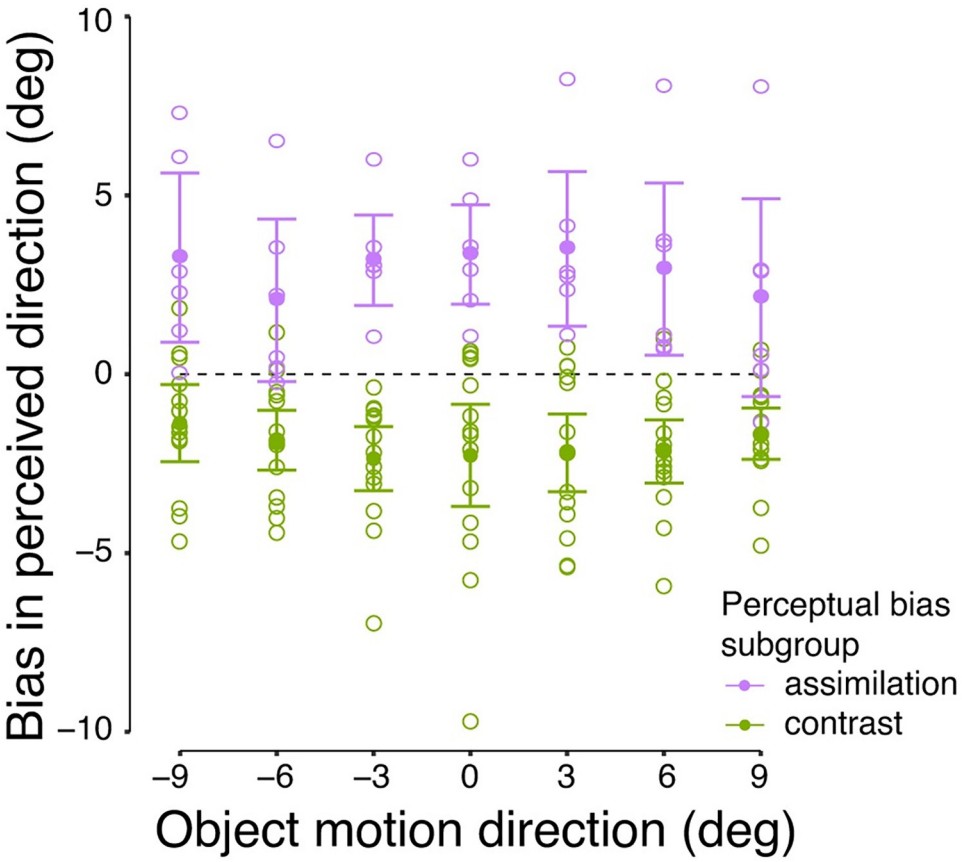

**Fig 8. Perceptual bias patterns in all observers.** Open symbols indicate the individual bootstrapped mean of perceptual bias in one object motion condition. Filled symbols indicate the average perceptual bias within each subgroup. Error bars indicate the 95% CI.

group showed the expected perceptual bias patterns with significant main effects of *internal motion* (assimilation: $F(1,5) = 9.05$, $p = 0.03$, $\eta_p^2 = 0.64$; contrast: $F(1,13) = 27.63$, $p < 0.001$, $\eta_p^2 = 0.68$). These findings indicate potential individual differences in perception that appear to be related to the strength of the pursuit bias.

Further exploratory analyses were performed to examine how pursuit biases between perceptual subgroups change over time, and how the net motion energy of retinal images change over time due to eye movements, see (S1 File). Motion energy provides an estimation of the neural activities encoding visual motion in the middle temporal area (area MT; [45,46]), and might indicate how perception could be affected. In general, we found that the difference in pursuit bias between perceptual subgroups only emerged in late steady-state pursuit (S1 Fig). In addition, net motion energy of retinal images during smooth pursuit was in the opposite direction to internal motion (S2 Fig). These findings provide additional insights in the interpretation of the across-observer correlation between perception and pursuit, and individual differences in perception, see Discussion.

## Discussion

In this study, we examined how perception and pursuit process motion signals within an object and integrate them with object motion signals. Whereas smooth pursuit eye movements consistently followed the average of internal and object motion, perceptual biases were

inconsistent across observers. Most observers showed a contrast bias in perception, but a few observers showed an assimilation bias. Interestingly, biases in perception and pursuit were correlated, showing that observers with a larger assimilation bias in pursuit tended to have a similar assimilation bias in perception. These results reveal a complex relationship between direction perception and smooth pursuit when processing motion signals at different spatial scales.

Different motion signals across space in the current study either associated with the whole object translating across the screen, or with internal object motion. This introduces ambiguity with regard to how to interpret the relationship between different motion signal sources. Our findings are in line with previous studies showing dissociations between perception and pursuit under similarly complex motion conditions. When perception and pursuit are faced with diverse motion signals from target and background [21] or from object and internal stripes [25], perception and pursuit are biased by background or internal motion in different ways. Whereas perception followed the relative difference between target and background speed, or showed no bias toward internal motion on average, pursuit always followed the vector average of all available motion signals. These different biases may reflect different functional demands of perception and pursuit.

Perception needs to recognize and classify visual information, thus segregating the target from its context [47,48]. Therefore, when the target is surrounded by a large-scale background, perception follows the relative speed between target and background [21]. When diverse motion signals occur all within the target, as in the current study, perception tends to show no bias overall. Observers in our study might not have consistently regarded internal motion as part of the scene from which the object motion had to be segregated. Individual differences in perceptual bias patterns might suggest that different observers either segregated or integrated object and internal motion (Fig 8). These differences are potentially due to different interpretations of whether the internal motion contributes to object motion or not. We will discuss additional reasons for individual differences in perception, and their implications, in the next section. Conversely, pursuit integrates all available motion signals in order to maximize the ability to monitor and collect information from the whole scene, resulting in an assimilation bias.

Studies have shown that different strategies—segregation and integration—can be used in parallel. When embedding a single, moving target dot within an RDK, the direction discrimination threshold of either the local (target dot) or the global (all background dots in the RDK) motion was not affected by whether observers were told whether to discriminate local or global motion [49]. This suggests that local and global motion directions were processed simultaneously, and that participants have access to both. During a dual task, in which observers were asked to attend to an individual dot within a group of dots while using their eyes to track the motion of the group, performance in both tasks was as good as when performing each task alone [50]. These studies indicate that segregation and integration can be performed simultaneously, potentially serving the different functional demands of perception and pursuit.

Our findings are different from studies showing an association between perception and pursuit when integrating ambiguous local motion signals [19,20]. In these studies, a line-figure object was only partially visible through an aperture, and observers were asked to track the object and report its motion direction. In this case, the local motion signals simply needed to be integrated to derive object motion. Therefore, unlike in the current study, there was neither a conflict between internal and object motion signals, nor ambiguity in how to interpret their relationship. Instead, both perception and pursuit relied on a coherent representation of the object motion, resulting in response similarity.

## Individual differences in perception, but not pursuit

An interesting dichotomy in perception and pursuit in the current study lies in the existence of individual differences in perception despite a consistent pursuit bias. Most observers (14 out of 20) showed a contrast bias, whereas some showed an assimilation bias. This individual difference in perception was also observed in a previous study comparing speed perception and pursuit velocity in response to translating and drifting gratings [25]. Congruently, Spering and Gegenfurtner [21] found larger variance in the correlation of perceptual responses with a model prediction of motion contrast than in the correlation of pursuit with an assimilation prediction. Below we will discuss two potential sources of ambiguity in the current study, which could drive the diversity in perception.

First, because we did not provide any instruction with regard to the internal motion in our stimuli, it is possible that observers had different interpretations of its relevance. Anecdotal conversations with observers suggest that some perceived the RDK as a ball rolling along a curved path, whereas others did not notice the internal dot motion. Therefore, some observers might have considered the internal motion as an intrinsic contributor to object motion, resulting in an assimilation bias, whereas others might have considered it as noise that should be either ignored or counteracted, resulting in a contrast bias. In order to track an object, the eyes need to integrate motion signals from all sources in order to continuously update motion information, and to respond to any potential object trajectory changes at short latency so that the object remains close to the fovea. Because the motion signals themselves are not ambiguous in the current study, pursuit averaged the available motion signals, mirroring results obtained in previous studies [21,24,25].

In addition to the uncertainty in the interpretation of internal motion, the ambiguity in motion energy in our stimuli could also contribute to a diverse perceptual response. Reverse motion perception has been observed widely with RDK stimuli, and occasionally with sinusoidal gratings. A small number of observers (usually less than 15%) may consistently perceive motion in the opposite direction to the actual dot motion [51]. Even with RDKs of 100% coherence, similar to the stimuli used in the current study, observers can still show reverse motion perception [52–54]. The probability of perceiving the actual motion direction presented in an RDK, or the opposite or orthogonal direction, corresponds to the relative strength of motion energy in all perceived directions [55]. Even RDKs of 100% coherence could contain motion energy in the opposite direction, which might contribute to the reverse motion perception [51]. The motion energy analysis for our stimuli shows that the net motion energy of retinal images along the vertical dimension was in the opposite direction to internal motion (S2 Fig), which could be a source of ambiguity in the direction signal. Although it is unlikely that the known reverse motion perception with RDK fully explains the individual differences observed here, it could contribute to task and stimulus uncertainty and enhance existing individual differences in perception.

In summary, several sources of uncertainty exist in the current task, including observers' interpretation of the task relevance of internal motion, and ambiguity in direction signals indicated by retinal motion energy. The need of the perceptual system to consider the context of the target might have resulted in discrepancies between observers in how object and internal motion should be processed. However, the pursuit system aims to monitor and collect information from the whole scene, thus performing signal integration irrespective of target-context interpretation.

## Correlation between biases in perceptual and pursuit responses

Notwithstanding the perceptual diversity and pursuit consistency, we found that biases in both responses were correlated across observers, but not on a trial-by-trial basis. This indicates that

even when perception and pursuit responses are linked, they likely have largely separated noise sources [56]. One possibility is that perception and pursuit share processing of low-level motion signals only at an early stage, for example, in early visual cortical areas such as the primary visual cortex (V1) and MT. However, previous studies [21,25] showing dissociations did not find correlations between perception and pursuit. Hughes [25] found that the correlation between speed perception and pursuit velocity across observers was not significant. Spering and Gegenfurtner [21] did not directly test the correlation between perception and pursuit across observers, but showed that speed perception and pursuit velocity were independently modulated by the relative direction of target and background motion, i.e., whether the background was moving in the same or opposite direction to target motion. Importantly, these two studies focused on speed perception, whereas the current study assessed direction perception.

That the exact motion feature under scrutiny matters is revealed by comparing previous studies that either reported trial-by-trial correlations between direction discrimination and pursuit [12–14], or did not report trial-by-trial correlations between speed discrimination and pursuit ([11,56]; but see [13,14]). Direction and speed information are processed together when observers view simple motion stimuli, such as sinusoidal gratings or random dots [57,58]. However, direction and speed discrimination differ depending on whether a stimulus moves along the cardinal axes or diagonally [59], and are learned independently through repeated practice of discrimination tasks [60]. Direction and speed information are also used differently when hitting moving targets in interceptive hand movement tasks [61]. In addition, single-pulse transcranial magnetic stimulation only impairs speed discrimination but not direction discrimination of moving stimuli [62]. These results indicate partially separate processing of direction and speed information [62,63], which might explain why the correlation between perception and pursuit across observers could differ depending on whether direction or speed was tested.

Another interpretation could be that interactions between perception and pursuit resulted in the correlation. For example, a top-down perceptual modulation on pursuit might contribute to the correlation between perception and pursuit. By performing further exploratory analyses of the temporal dynamics in pursuit biases between perceptual subgroups (Supporting Information), we found that the differences in pursuit bias between subgroups only emerged in late steady-state pursuit, whereas both groups have similar pursuit biases during early pursuit phases. In addition, during the late steady-state phase, some observers from the contrast group even showed a pursuit bias in the opposite direction to internal motion (S1B Fig), similar to their perceptual biases. It has been suggested that a perceptual decision could be formed during the stimulus display once enough evidence has been accumulated. When a perturbation of motion coherence of the RDK stimulus was introduced at a random time within a trial, perceived direction was biased more by the coherence perturbation when it was presented earlier than later in the trial [46]. By comparing discrimination thresholds from data and theoretical discrimination thresholds calculated based on the assumption of perfect integration of all stimulus information presented, the authors proposed that the perceptual system might have made a decision earlier during stimulus display (about 400 ms after stimulus onset in this task) and not fully utilized the later stimulus information [46]. Therefore, it is possible that in the current study later pursuit was more aligned with perception, once the perceptual decision was formed.

## Limitations and future directions

In this study, we showed different but related perception and pursuit responses to object vs. internal motion. The exact mechanisms of how perception and pursuit process diverse motion

signals remain unclear. One confound that might have affected perceptual bias is the final RDK location. Although we emphasized to observers that using the last location of the RDK would not be beneficial, it is not guaranteed that observers strictly followed the instruction. In future studies, it is important to ensure that perceptual reports are not contaminated by any potential location effects.

Besides the potential effect of location bias, it remains unclear whether individual differences in patterns of perceptual bias are related to different interpretations of internal motion, or stimulus properties such as motion energy. Different manipulations of stimuli and tasks are required to confirm these speculations. One manipulation could be the use of more realistic stimuli without ambiguity in the relationship between different motion signals, e.g., a flying and rotating volleyball (Fig 1B). If ambiguity mediates the integration of internal and object motion, we would expect smaller diversity in perception with stimuli that have little ambiguity in how object and internal motion are related.

Moreover, stimulus parameters could be manipulated to present conditions with different net motion energies. If the ambiguity in direction signals indicated by motion energy contributes to the individual differences in perception, we should observe similar patterns of the perceptual bias with stimuli with strong net motion energy in the target direction. It would also be interesting to see whether pursuit bias is affected by these manipulations. If so, it would imply that perception and pursuit are affected by similar top-down modulations.

To examine whether the across-observer correlation between perception and pursuit is caused by shared motion processing or by a top-down perceptual modulation on pursuit, examination of the temporal dynamics in the relationship between perception and pursuit is needed. One limitation of the current study is that perception was only measured at the end of the trial, whereas eye movements were recorded continuously throughout the trial. As a result, observers could have had more processing time for the perceptual response than for the pursuit response. In addition, because we did not restrict the time for observers to respond, there was a long reaction time (about 2 s) between the display offset and the perceptual report. The difference in the available processing time between perceptual and pursuit measurements could contribute to their dissociation, even though perception and pursuit share processing of motion signals [64]. To address the issue of different temporary limitations when measuring pursuit and perceptual responses, future studies could probe perception at different time points and compare it to pursuit at corresponding time points. If motion processing is shared between perception and pursuit, we would expect perception to have similar temporal dynamics as pursuit (i.e., change of bias over time). In addition, we would expect a correlation between perception and pursuit biases with short display duration. If a correlation between perception and pursuit biases only exists when the display duration is long, it might be driven by a top-down perceptual modulation on pursuit rather than by shared low-level motion processing between perception and pursuit.

## Conclusions

Using RDKs with translating object motion and additional internal dot motion, we showed that smooth pursuit to object motion is biased in the direction of internal motion, whereas perception was unbiased on average, with striking individual differences between observers. Different functional demands of the two systems might have resulted in the dissociation between perception and pursuit when processing motion signals at different spatial scales. The overall lack of bias in perception and the potential contrast bias in perception found in more than half of the observers are consistent with the need of perception to perform object segregation from the scene. Conversely, pursuit performs signal integration and follows the average

motion signal to maximize visual information collection within the scene. Signal segregation and integration may reflect dependencies on different MT neurons with either inhibitory or excitatory center-surround receptive fields. Interestingly, the correlation between perceptual and pursuit biases indicates a potential link in early-stage processing and/or interactions between the two systems.

## Supporting information

**S1 Fig. Biases in perceptual subgroups over time.** (a) Color indicates the perceptual bias subgroup, see legends in panel (b). Solid lines indicate the mean pursuit bias in each subgroup. Shaded areas indicate the 95% CI. Dashed vertical lines indicate time points of the pursuit onset, and the start, middle point, and end of the steady-state phase analysis window. (b) Biases in pursuit direction in perceptual subgroups across the three pursuit phases. Horizontal bars indicate the mean across observers. Error bars indicate the 95% CI. Circles indicate the mean of individual observers.
(TIF)

**S2 Fig. Biases in net motion energy in the vertical dimension between perceptual subgroups.** Positive values indicate that there was more motion energy in the same direction as internal motion. Solid lines indicate the mean of each perceptual subgroup. Shaded areas indicate the 95% CI. Dashed vertical lines indicate time points of the pursuit onset, and the start, middle point, and end of the steady-state phase analysis window.
(TIF)

**S1 File. Further exploratory analyses between perceptual subgroups.** This file provides details on the further exploratory analyses conducted. These analyses include the comparison of pursuit biases between perceptual subgroups over time, and how motion energy of retinal images changes over time in the perceptual subgroups.
(PDF)

## Acknowledgments

We thank the members of the Spering lab, especially Philipp Kreyenmeier and Doris Chow, for their feedback on data analysis and interpretation, and comments on the manuscript.

## Author Contributions

**Conceptualization:** Xiuyun Wu, Miriam Spering.

**Data curation:** Xiuyun Wu.

**Formal analysis:** Xiuyun Wu.

**Funding acquisition:** Miriam Spering.

**Investigation:** Xiuyun Wu.

**Methodology:** Xiuyun Wu, Miriam Spering.

**Project administration:** Xiuyun Wu, Miriam Spering.

**Resources:** Miriam Spering.

**Software:** Xiuyun Wu.

**Supervision:** Miriam Spering.

**Validation:** Xiuyun Wu.

**Visualization:** Xiuyun Wu.

**Writing – original draft:** Xiuyun Wu.

**Writing – review & editing:** Xiuyun Wu, Miriam Spering.

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
