## [Decision Letter · Decision Letter 0]

15 Jul 2022

PONE-D-22-14489Tracking and perceiving diverse motion signals: Directional biases in human smooth pursuit and perceptionPLOS ONE

Dear Dr. Wu,

Thank you for submitting your manuscript to PLOS ONE. After careful consideration, we feel that it has merit but does not fully meet PLOS ONE’s publication criteria as it currently stands. Therefore, we invite you to submit a revised version of the manuscript that addresses the points raised during the review process. The two reviewers found only minor comments to make to your manuscript, as you will see below. Please, read these comments carefully and address them into your revised version. I am fully confident that this next version will answer all the remarks of the reviewers.

We look forward to receiving your revised manuscript.

Kind regards,

Robin Baurès, Ph.D.

Academic Editor

PLOS ONE

Journal Requirements:

Reviewers' comments:

Reviewer's Responses to Questions

**Comments to the Author**

1. Is the manuscript technically sound, and do the data support the conclusions?

Reviewer #1: Yes

Reviewer #2: Yes

2. Has the statistical analysis been performed appropriately and rigorously? 

Reviewer #1: Yes

Reviewer #2: Yes

3. Have the authors made all data underlying the findings in their manuscript fully available?

Reviewer #1: Yes

Reviewer #2: Yes

4. Is the manuscript presented in an intelligible fashion and written in standard English?

Reviewer #1: Yes

Reviewer #2: Yes

5. Review Comments to the Author

Reviewer #1: This work attempted to determine whether pursuit and perception integrate object motion with internal object motion. The authors found that smooth pursuit eye movements relative to object motion were biased in the direction of internal motion, but motion perception was unbiased on average, with striking individual differences between observers. In addition, the perceptual bias was related to the magnitude of the pursuit bias. They interpret that the dissociations between perception and pursuit may reflect the different functional demands of the two systems.

Overall, this is a nice study that is well written and has interesting results. The data are presented properly and support the conclusions. These findings are important and valuable for our understanding of how perception and pursuit are related and interact with each other. I have a few questions/suggestions for improvements to the manuscript (specific comments).

Specific comments:

1. Regarding the perception task, please provide how long it took observers to respond. If there is a time gap between pursuit and perception responses, is it possible that this is related to the difference in bias?

2. According to the Methods, the observers were encouraged to focus on the angle of the object's motion trajectory rather than referring to the last RDK position to determine the direction of the object. This is understandable, but I wonder how it can be guaranteed that the observer is not simply using the final position of the RDK as a cue. Please clarify.

3. In the Results, the authors describe that "In contrast to the consistent bias in the average motion direction in pursuit, across all observers, perceived direction was not affected by internal motion direction (Fig 6a)." However, since the effect of internal motion on perception varies in each observer, the expression "not affected" is misleading, thus it is recommended to correct the description.

4. It is not clear whether the pattern "a" (assimilation) or "b" (contrast) in figure 6 depends on the influence of attention during the experiment, e.g., attention to internal motion, or attention to global motion, as described in the discussion. So, it would be helpful for readers if the author could further explain what factors determine each pattern (assimilation or contrast).

5. The authors described that "Different biases in perception and pursuit might reflect dependency on different types of MT neurons." However, if observers were similarly exposed to visual stimuli, one would assume that MTs would show similar activity because similar retinal slip would occur from RDKs. Nevertheless, please explain additionally how the results of the significant inter-observer individual differences in perception can be understood from the MT activity. It is difficult to understand from the reader's point of view that the difference biases between perception and pursuit could be based on different MT activities.

Reviewer #2: The current manuscript uses complex stimuli that are moving in the environment, but that also contain internal motion, and asks how human participants make eye movements (smooth pursuit) to these stimuli, as well as asking about people’s subjective experience of them. The authors find that smooth pursuit eye movements are biased in the direction of internal motion, whereas there was more variability in perceptual responses: however, there was still a correlation between perception and pursuit, with participants showing a stronger pursuit bias being more likely to report a directionally aligned perceptual bias. The manuscript is well written and clear, and the analyses seem to be done correctly and interpreted appropriately. The authors have also done an excellent job of making their data and code available, with very comprehensive ‘read me’ files. I have only a few minor comments that might help to improve the manuscript.

L130 – ‘overall link between perception and pursuit’ might sound better.

L193 – I wonder if there is any way to check if some participants did use the last location of the RDK as a reference, and whether this explains any of the variability in perceptual response?

L218 – can you give a ballpark figure for how frequently trials had saccades in them?

L240 – it might be worth being very clear here about what the dependent variables are – I think you are running 2 repeated measures ANOVAs, one with the directional bias for pursuit and the other with the directional bias for perception, but this isn’t actually spelled out explicitly.

Fig 3 – having predictions in a graph like this is very helpful, thank you! A very minor point is that the colours don’t work very well in black and white (but probably not that important as this is an online only journal).

L262 – thinking about how you specified this model – is there an argument for including any interactions? I can see why you haven’t (as you don’t find e.g. a significant interaction between object motion and internal motion in your perceptual rmANOVA) but this seems slightly like hypothesising after the fact given that you included the interactions in those original rmANOVAs. Similarly, Fig 4C/5C seems to suggest that you might expect pursuit bias to interact with object motion direction and internal motion direction, so it might be sensible to include the full set of interactions here.

Fig 7 – is there any way to indicate the confidence interval on this figure?

6. PLOS authors have the option to publish the peer review history of their article (what does this mean?). If published, this will include your full peer review and any attached files.

Reviewer #1: No

Reviewer #2: **Yes: **Anna Hughes

---

## [Author Response · Author response to Decision Letter 0]

31 Aug 2022

We thank the two reviewers and the editor for their constructive comments and positive feedback. All issues raised by the reviewers have been addressed point-by-point in the Rebuttal letter.

---

## [Decision Letter · Decision Letter 1]

14 Sep 2022

Tracking and perceiving diverse motion signals: Directional biases in human smooth pursuit and perception

PONE-D-22-14489R1

Dear Dr. Wu,

We’re pleased to inform you that your manuscript has been judged scientifically suitable for publication and will be formally accepted for publication once it meets all outstanding technical requirements.

Kind regards,

Robin Baurès, Ph.D.

Academic Editor

PLOS ONE

Reviewers' comments:

Reviewer's Responses to Questions

**Comments to the Author**

1. If the authors have adequately addressed your comments raised in a previous round of review and you feel that this manuscript is now acceptable for publication, you may indicate that here to bypass the “Comments to the Author” section, enter your conflict of interest statement in the “Confidential to Editor” section, and submit your "Accept" recommendation.

Reviewer #1: All comments have been addressed

Reviewer #2: All comments have been addressed

2. Is the manuscript technically sound, and do the data support the conclusions?

Reviewer #1: Yes

Reviewer #2: Yes

3. Has the statistical analysis been performed appropriately and rigorously? 

Reviewer #1: Yes

Reviewer #2: Yes

4. Have the authors made all data underlying the findings in their manuscript fully available?

Reviewer #1: Yes

Reviewer #2: Yes

5. Is the manuscript presented in an intelligible fashion and written in standard English?

Reviewer #1: Yes

Reviewer #2: Yes

6. Review Comments to the Author

Reviewer #1: The authors have addressed all comments and revised the manuscript appropriately. I have no further comments to make.

Reviewer #2: The authors have addressed all my comments and I have no further concerns. I recommend publication of the manuscript.

7. PLOS authors have the option to publish the peer review history of their article (what does this mean?). If published, this will include your full peer review and any attached files.

Reviewer #1: No

Reviewer #2: No

---

## [Editor Report · Acceptance letter]

19 Sep 2022

PONE-D-22-14489R1 

Tracking and perceiving diverse motion signals:
Directional biases in human smooth pursuit and perception 

Dear Dr. Wu:

I'm pleased to inform you that your manuscript has been deemed suitable for publication in PLOS ONE. Congratulations! Your manuscript is now with our production department. 

Kind regards, 

on behalf of

Dr. Robin Baurès 

Academic Editor

PLOS ONE